# The impact of the internationalization of China's new retail industry on corporate performance—A moderating effect based on proprietary assets

Li-Wei Lin[1], Shih-Yung Wei[2]*

1 College of Business Administration, Fujian Jiangxia University, Fuzhou City, Fujian, China, 2 Business School of Yulin Normal University, Yulin City, Guangxi, China

* 2057085290@qq.com

## Abstract

### Purpose

The purpose of this study is to examine the factors influencing China's new retail industry on corporate performance. We mainly design the adjustment effect of the internationalization of its new retail industry on corporate performance and proprietary assets.

### Design/Methodology/Approach

The framework is based on dynamic panel data of 590 listed companies in China's new retail industry from 2007 to 2019.

### Findings

We apply the concept of big data for the analysis and investigation, including the DOI explanatory variable for the degree of internationalization, Tobin's Q explained variable, and adjusted variable of proprietary assets (R&D intensity RDI, marketing intensity MI, and capital intensity CI). We found that the degree of internationalization of the new retail industry has a positive impact on corporate performance.

### Research limitations/implications

The resultant findings only reflect the setting of China's new retail marketplace. With the research model developed here in, future research can target cross-country studies of various forms of online and offline market sites to determine regional differences in the development of new retail activities.

### Originality/Value

The results show that the major new retail industry is a multi-channel concept that affects overall corporate performance. The authors focus on corporate performance, which is a moderating effect based on proprietary assets.

**Data Availability Statement:** All relevant data can be found in the supporting information files or from China's NetEase. http://quotes.money.163.com/stock

**Funding:** We don't have any source of funds.

**Competing interests:** We have no conflicts of interest.

## 1. Introduction

The new retail industry is important to internationalization, and China's new retail model can serve as a reference for other countries, making it an interesting topic for research. We thus combined the concepts of marketing, financial management, and econometric methods to break through the traditional concept of only pure economics. This study mainly investigated data of China's new retail industry and analyzed listed companies from 2007 to 2019. We aggregated the concepts of the supply chain, finance, big data, and econometrics to conduct this survey and analysis. The new retail industry is a form of cooperation and applied to stores mainly through virtual reality (VR) or augmented reality (AR). In addition, the concept of 5G can also be combined to accelerate the internationalization of the new retail industry and improve its performance. China has fully laid out its 5G plan in 2020, which can further improve the internationalization and performance of the new retail industry.

Corporate performance is important for new retail companies, especially in the face of the COVID-19 epidemic, which has caused many industries to suffer in terms of performance. In the face of environmental instability, the new retail industry needs to increase the transformation of internationalization into corporate performance. Lin et al. (2022) referred to Intelligent Capital, Organizational Learning, and Corporate Performance Influence Relationship [1]. We know that the new retail industry uses the sharing of knowledge or information, but such a series of intangible asset processes are often overlooked as part of corporate performance.

Tobin's Q refers to the difference between the market value and book value of tangible assets. This means that many companies only see tangible assets when observing their own performance, but forget the concept of intangible assets. Why do we choose a new retail industry to observe and analyze? In the operation process of a major new retail sector, there will be brand support, especially the value of brand building in many new retail industries in China. However, this invisible value is often ignored in terms of its intangible asset value. Damodaran (2002) mentioned the concept of investment evaluation and explained the relationship between investment assets through the concept of Tobin's Q [2]. Few people have paid attention to and evaluated the market value in the new retail industry, which leads to inaccuracy when predicting the performance of the whole new retail industry. The difference between market value and book value is also ignored and applied by many listed companies in their financial statements.

Why should one explore the impact of internationalization on corporate performance in this new retail industry? The most important thing is that in the operation of new retail products, the value created by many online brands is often ignored with in corporate performance measurement. This paper thus applies the theory of social capital transaction to analyze and discuss the transaction between buyers and sellers of the new retail industry, so that the overall new retail industry can present its performance in transactions and further exhibit corporate performance from tangible assets to intangible brand assets. The social capital theory is a very important theoretical concept in economics. Physical capital and human capital both affect the overall direction of operations and further impact overall organizational performance.

Fukuyama (1999) mentioned that the social capital theory mainly enables members within a company to cooperate with each other to further create the performance and value of the company [3]. Timothy and Toni (2012) proposed that Tobin's Q is one of the most frequently measured financial indicators, including the error term that is generated in the measurement process [4]. These are all regression correlations that need to be analyzed in the measurement process.

China's well-known One Belt and One Road is a grand internationalization policy. So far, no one has studied such a topic in regards to the internationalization of the new retail industry.

**Table 1. Illustrative research summarizing the antecedents to corporate performance.**

| Illustrative research | Context | Theoretical basis | Antecedents to performance in corporate performance | Key findings or propositions |
|---|---|---|---|---|
| HENGCHEN DAI et al. (2018) [5] | Others' expectations to manage impressions | self-expectations theory | Affect, Impression | The findings suggest that affect and External Expectations and Impression Management impact performance. |
| Ehsan Moradi et al. (2021) [6] | Between business model innovation and open innovation | Decision Theory | Effective organizational innovation is the key to make and keep a competitive advantage | The findings suggest that innovation suggests that companies and businesses should be more flexible and open to innovation processes. |
| Elfadil A. Mohamed et al. (2020) [8] | Impact of corporate performance on stock price predictions | Expectations theory | Intelligence, GAs, sentiment analysis, and deep learning | The findings suggest that used a fused model of an ANN with PCA for forecasting stock prices on the Tehran Stock Exchange. |
| Fisher (1930) [7] | Long-run dynamic adjustments of the term structure of interest rates | expectation theory | Interest rates, dynamic adjustment | The results show that s investors' expectations about future spot interest rates affect current long-term interest rates. |
| Liping Qian et al. (2018) [9] | Between online sellers and buyers | Social capital theory | Trust | The results show that Hindering or enabling structural social capital to enhance buyer performance? |
| Heinz-Theo Wagner et al. (2014) [10] | Between online sellers and buyers Value Co-Creation | Social capital theory | IT alignment and business value | The findings suggest that it drives operational alignment and IT business value. |
| SHINTARO OKAZAK et al. (2017) [11] | Between social media knowledge sharing | Social capital theory | Knowledge Sharing | The findings suggest that Knowledge Sharing Social Media |

We break up this dilemma to follow such a research direction. Unceasingly the new retail industry in China to other countries, including the listed on alibaba to instruction in southeast Asia and Hong Kong, QQ the internationalization of investment, and shrimp, are all in copy China's successful e-commerce business model to other countries, and further to achieve the new retail industry internationalization, and the corporate performance directly and indirectly reflected in the financial statements. The continuous internationalization of the new retail industry and the creation of brand value are often ignored by many enterprises, resulting in the failure of immediate and positive corporate performance.

In the process of internationalization investment, sometimes the end results do not appear immediately, and long-term investment costs do arise. This is why many large enterprises will consider the investment costs of internationalization during the process of operations. Our research objective is to observe the adjustment effect of the internationalization of China's new retail industry on corporate performance assets, which is an interesting and valuable research design. China's new retail industry is expanding its internationalization, making it worth studying the topic of China's overall retail industry. For the degree of internationalization, few people in the academic community pay attention to it, but it can be calculated through empirical mathematics. From such a research method, the statistical analysis offers a precise and significant effect. In Table 1 we directly see the basic theories and findings concerning the topic of corporate performance [5–11].

## 2. Theoretical analysis and research hypothesis

### 2.1 Corporate performance

Operating performance refers to a company's operating efficiency and operations performance during a certain time period. A company's operating efficiency level is mainly reflected in its profitability, level of asset operations, solvency, and follow-up development capabilities. Wei-Kang Wanga (2021) noted the Management characteristics and corporate performance of

Chinese chemical companies [12]. Yasas L. Pathiranage (2020) offered a literature review on organizational culture towards corporate performance [13].

Corporate performance attaches great importance to the operation of an enterprise and its financial statements. One can take China's new retail industry as an example. In the Internet era of competition, the overall new retail industry pays more attention to corporate performance. Why is Tobin's Q used to analyze and observe corporate performance in our study? One of the most important reasons is that there are many intangible assets in the new retail industry that do not appear in financial statements, which is why our research fills the gap in the literature.

Timothy Erickson et al. (2012) proposed that Tobin's Q is a common measurement index and tool for evaluating corporate performance [14]. Reed (2010) [15], Shaheen (2012) [16], and Ernayani and Robiyanto (2016) put forward that profitability has a significant impact on corporate performance [17]. Corporate performance often neglects a firm's market value and some intangible assets, which may lead to the loss of some corporate performance in its financial statement. Many scholars use ROA and ROE to analyze corporate performance, but we directly use Tobin's Q as the explained variable of corporate performance.

## 2.2 Degree of internationalization

Hitt et al. (2006) pointed out that internationalization refers to the various operational activities adopted by manufacturers in expanding the sales of their products or services to transnational markets and transnational regions [18]—that is, sales, R&D, manufacturing, and other operational activities done outside the home country can be called internationalization. Ziyi Wei et al. (2019) presented Chinese service multinationals' impact on the degree of internationalization and performance [19]. Carlos González (2019) spoke of DOI by the extent of a firm's top managers' international experience and the amount and cultural zone dispersion of the firm's transnational board interlocking [20]. Leandro Rodrigo Canto Bonfim et al. (2018) examined managers' international cognition and how risks affect company performance [21].

The degree of internationalization is an interesting variable that is rarely used as an explanatory variable in financial management. When foreign sales account for total sales or foreign sales exceed total assets, societic-based internationalization indicators show less foreign resource interdependence visible throughout a company's value chain. The concept of the social capital theory can be applied to the internationalization of the new retail industry. Through the mutual transaction relationship between buyer and seller, the resource needs of both parties can be further reached and gradually expanded to the degree of internationalization. Sullivan (1994) mentioned that the degree of internationalization has an impact on corporate performance [22]. Many of the new retail industries have gone to the overseas market. With a diverse mix of international experience and management models, senior executives with international experience and cultural districts are linked to a company's multinational board of directors. We use Alibaba and Tencent QQ to evaluate their market expansion in Southeast Asia. Their degrees of internationalization will have a certain impact on their performance in the future.

Oesterle and Richta (2013) proposed the influence of DOI on the degree of internationalization [23]. They designed and conducted stability detection and empirical analysis for some researchers. We use exploratory and confirmatory factor analysis to reveal and test the multidimensional structure of DOI construction and investigate the factor relationship between them. Hitt et al. (2006) set up 25 different DOI architecture operation modes. The operation of internationalization directly affects overall corporate performance, which can be explained from the enterprise resource theory [24]. Resource-based theory (RBT) mainly discusses the

resource needs in economics to be applied to the business activities of companies and to create their corporate performance. In terms of the degree of internationalization of the new retail industry, a company can continue to expand into foreign markets through e-commerce platforms. These business activities will indirectly create the brand value and intangible assets of the company, and such continuous creative activities will affect company performance. Johanson and Vahlne (1977) mentioned that the degree of internationalization refers to the accumulation of operating experience in foreign markets [25].

### 2.3 Proprietary assets

Proprietary assets refer to the ratio of shareholders' equity (capital plus profits) to assets, also known as self-owned capital ratio and property right ratio. They correspond to the debt ratio, which refers to the ratio of liabilities to assets. Jin K. IHan et al. (2001) presented capital requirements, cost advantages, switching costs, distribution access, and proprietary assets [26]. In this paper, we investigate and analyze proprietary assets (R&D intensity (RDI), marketing intensity (MI), and capital intensity (CI)). Our primary objective is to determine whether the RDI, MI, and CI of a proprietary asset will influence the relationship between cause and effect variables in the overall architectural design.

The degree of internationalization of China's new retail industry is mainly due to the fact that this industry as a whole has to go abroad. The new retail industry as a whole needs to observe whether it is affected by internal and external factors of some companies, which affect overall corporate performance. For many industries, few scholars have considered the application of proprietary assets to observe and analyze the overall regulatory variables. Proprietary assets are an interesting variable, but many companies' financial statements do not analyze and view them in this way. The degree of internationalization can transform the competitive advantage of the new retail industry in foreign markets through proprietary assets.

How can the new retail industry transform itself into competitiveness in 2020 through globalization, newer asset capabilities, and approaches? Barros et al. (2005) mentioned that some European banks have increased their asset capacity through cross-border mergers and acquisitions [27]. Almeida (1996) proposed that mnes increasingly value their expertise and capabilities in overseas markets. Thus, we can see that the internationalization of China's new retail industry in overseas markets will enhance its proprietary assets [28]. Wei and Lin et al. (2019) stated that proprietary assets have an impact on corporate performance [29]. Wei and Lin et al. (2020) pointed out that corporate assets have an interactive effect on performance [30]. We thus offer the first 2 hypotheses.

Hypothesis 1A: The degree of internationalization has a positive impact on firm performance.

Hypothesis 2A: Proprietary assets have a positive impact on company performance.

### 3. Study design

We used the cabinet information of The Oriental Fortune website in China. With such unstable panel data, we applied the programming method to pick out the missing values one by one and took the panel data into consideration without rigor and circumference. We sort out and analyze the data through the instability of the panel data, and data analysis and application description will be done later.

This study selected a total of 58 Chinese new retail industry stocks (the east wealth network, http://data.eastmoney.com/) for the period 2007–2019. There were some unlisted companies during the study period. There is also a lack of data or data without the company's sales

regions, and so the sample has a total of 590 observations. This type of research belongs to the panel data of unbalanced information.

This study examines the effect of the degree of internationalization on corporate performance and the effect of the regulation of asset specificity. Therefore, variables were divided into four factors: the explained variable degree of internationalization (DOI), the moderating asset specificity RDI (research and development strength, marketing strength, MI CI) and capital strength, and the explanatory variable of corporate performance (Tobin's Q) CFP. In order to solve the explained variables of endogenous and exogenous problems, this research has five control variable indicators: company size (SC), the proportion of property rights (ER), stock pledge proportion (PL) of major shareholders, power issues (chairman and general manager) (BM), and the length of listing time (AG).

### 3.1 Variables' explanation

There are many indicators for the degree of internationalization. This study used the sales region (Michel and Shaked (1986) [31]; Shaked (1986) [32]; Wei et al. (2019) as a measuring variable, in order to make the variable converge with those of a normal distribution [33]. For numerical transformation, we take the natural logarithm of overseas sales divided by the domestic sales (each plus one avoid is zero ability to calculate natural logarithm), calculated as follows (source from the annual reports published in revenue, will the classification of domestic and overseas sales and then through the above formula):

$$DOI = \mathrm{LN}\left(\frac{\mathrm{OS}+1}{\mathrm{WS}+1}\right) \qquad (1)$$

OS: Percentage of overseas sales
WS: Percentage of domestic sales

### 3.2 Moderating variable

It was first proposed by Morck and Yeung (1991) that the influence of the degree of internationalization on corporate performance should include proprietary assets as the regulatory variable [34]. However, there are many definitions of proprietary assets. In this study, Jung (1991) used research and development intensity, marketing intensity, and capital intensity as indicators [35].

$$\mathrm{RDI}(t)_i = \frac{\mathrm{RD}(t)_i}{\mathrm{S}(t)_i} \qquad (2)$$

RD$(t)_i$: Company i's R&D expenses in phase t
S$(t)_i$: Company i's revenue in phase t

$$\mathrm{MI}(t)_i = \frac{\mathrm{MK}(t)_i}{\mathrm{S}(t)_i} \qquad (3)$$

MK$(t)_i$: Company i's R&D expenses in phase t
S$(t)_i$: Company' is marketing expenses in phase t

$$\mathrm{CI}(t)_i = \mathrm{LN}(\frac{\mathrm{LA}(t)_i}{\mathrm{LA}(t-1)_i})) \qquad (4)$$

LA$(t)_i$: Company i's non-current assets in phase t
Source: Annual reports of major companies and calculations in this study.

## 3.3 Explained variable

ROA, ROE, and Tobin's Q are often used in the application of corporate performance because ROA and ROE are accounting calculation values and are often beautified by windowing effects. Therefore, market performance (namely, Tobin's Q) is proposed to be adopted for corporate performance in this study. However, the debt decoration quality of Tobin's Q is difficult to obtain, and so Proxy Q proposed by La Porta et al. (2002) is proposed herein [36].

$$\mathrm{CFP(t)_i = Proxy\ Q(t)_i = \frac{ME(t)_i + BD(t)_i}{BA(t)_i}} \tag{5}$$

ME(t)$_i$: Market value of company i in phase t (common stock + preferred Stock)
BD(t)$_i$: Company i's total book liabilities in phase t
BA(t)$_i$: Company i's total book assets in phase t
Sources: Shanghai and Shenzhen Stock Exchange (share prices), annual reports of major companies, and calculations in this study,

## 3.4 Control variables

Studies of corporate performance are almost all on the company scale, where the proportion of property rights and the length of time to market are control variables. Company scale considers economies of scale, and property in proportion to the capital structure variables (Myers, 1977 confirmed that the earliest) has an effect on corporate performance. According to the characteristics of Chinese securities laws, we use the enterprise's listing's length of time as a control variable [37].

The variables related to corporate governance are replaced by two indicators, Chairman and General Manager respectively (Jensen, 1993; Yermack, 1996) [38] and stock pledge status of major shareholders (Yeh and Lee, 2001) [39].

$$\mathrm{Company\ size = LN\ (total\ assets)} \tag{6}$$

$$\mathrm{ER(t)_i = LN\left(\frac{BD(t)_i}{BE(t)_i}\right))} \tag{7}$$

BE(t)$_i$: Company i's total book equity in phase t

$$\mathrm{BM = \begin{cases} 1 & \mathit{Chairman}\ \text{and general manager} \\ 0 & \text{other} \end{cases}} \tag{8}$$

$$\mathrm{PL = \frac{\mathit{Quantity}\ \text{of pledge by majority shareholding}}{\text{Total}\ \mathit{quantity}\ \text{of majority shareholding}}} \tag{9}$$

Length of listing (AG)

$$\mathrm{AG(t)_i = LnDATA(t)_i - IPO_i} \tag{10}$$

DATA(t)$_i$: Company i in the current period
IPO$_i$: Company i's listing time
Through the introduction of the variables above, the index statistics of all variables are shown in Table 1 below.

From Table 1, new sales are given priority in the domestic retail industry (DOI < 0), DOI = 3.99), said 2% of export for sale in the domestic market, while the value of the company

**Table 2. The variable narrative statistical scale was studied.**

|         | CFP   | DOI   | RDI  | MI   | CI    | SC    | ER    | BM   | PL   | AG    |
|---------|-------|-------|------|------|-------|-------|-------|------|------|-------|
| Obs.    | 590   | 590   | 590  | 590  | 590   | 590   | 590   | 590  | 590  | 590   |
| Mean    | 2.17  | -3.99 | 0.01 | 0.12 | 0.16  | 22.35 | 0.02  | 0.20 | 0.18 | 8.11  |
| Med.    | 1.69  | -4.62 | 0.00 | 0.09 | 0.09  | 22.27 | 0.15  | 0.00 | 0.02 | 8.51  |
| Max.    | 12.48 | 1.59  | 0.21 | 0.52 | 2.66  | 26.19 | 4.67  | 1.00 | 0.94 | 9.20  |
| Min.    | 0.63  | -4.62 | 0.00 | 0.00 | -1.38 | 18.88 | -3.12 | 0.00 | 0.00 | 2.20  |
| Std. D. | 1.56  | 1.26  | 0.02 | 0.09 | 0.32  | 1.11  | 0.96  | 0.40 | 0.24 | 1.08  |
| Sk      | 3.28  | 2.12  | 4.25 | 1.47 | 2.54  | 0.47  | -0.18 | 1.53 | 1.34 | -2.00 |
| K       | 17.26 | 6.60  | 25.20| 5.86 | 17.62 | 3.24  | 3.92  | 3.33 | 3.70 | 8.02  |

of the new retail industry is affected by the market, the average is greater than 2 (2.17), the cost of r&d ratio (0.01) is far less than the cost of marketing percentage (0.12), the growth of the company's assets with an average of 16% a year, the chairman and the mechanism of total accounted for about 2, big shareholders equity pledge proportion is 18%, data distribution, in addition to property ratio (0.18), the length of time to market (2) as the left. The other variables are right-biased. All the data show a high gorge peak (K>3), especially R&D intensity of 25.2, indicating that some companies invest much more in R&D than other companies (the maximum is 0.21).

## 4. Research model

Based on the introduction of the above research variables, the model of this study is established as follows.

$$CFP = \beta_0 + \beta_1 DOI + \beta_2 DOI \cdot + \beta_3 DOI^3 + \beta_4 RDI + \beta_5 MI + \beta_6 CI$$
$$+ \beta_7 SC + \beta_8 SC^2 + \beta_9 ER + \beta_{10} PL + \beta_{11} BM + \beta_{12} AG$$

Using higher powers is likely to cause collinear problems among variables (variables are highly correlated), as shown in Table 2. Therefore, this study changed the model to decentralize the data so as to solve the collinear problem. After decentralization, the relationship between the higher power and the higher power of internationalization was solved, as shown in Table 3. The final model is as follows.

$$CFP = \beta_0 + \beta_1 DOI + \beta_2 DOI \cdot (DOI - \overline{DOI}) + \beta_3 DOI \cdot (DOI - \overline{DOI})^2 + \beta_4 RDI + \beta_5 MI + \beta_6 CI$$
$$+ \beta_7 SC + \beta_8 (SC - \overline{SC})^2 + \beta_9 ER + \beta_{10} PL + \beta_{11} BM + \beta_{12} AG$$

Table 3 shows that there is no high phase between variables, but whether there is a collinearity problem. This study uses the Engle and Granger (1987) co-integration test [40]. The test result T-statistic = -5.671322, p≤0.001, indicating that there is no cointegration.

The relationship between each explanatory variable and the explained variable can be illustrated by Table 4. This study found that the square term of DOI (-0.13), SC (-0.44), ER (-0.43),

**Table 3. Internationalization degree high power correlation coefficient matrix.**

|      | DOI   | DOI2  | DOI3 |
|------|-------|-------|------|
| DOI  | 1     |       |      |
| DOI2 | -0.97 | 1     |      |
| DOI3 | 0.94  | -0.99 | 1    |

**Table 4. Study variable correlation coefficient matrix table.**

| | CFP | DOI | $DOI \cdot (DOI-\overline{DOI})$ | $DOI \cdot (DOI-\overline{DOI})^2$ | RDI | MI | CI | SC | $(SC-\overline{SC})^2$ | ER | BL | BM | AG |
|---|---|---|---|---|---|---|---|---|---|---|---|---|---|
| CFP | 1 | | | | | | | | | | | | |
| DOI | 0.10 | 1 | | | | | | | | | | | |
| $DOI \cdot (DOI-\overline{DOI})$ | -0.13 | -0.70 | 1 | | | | | | | | | | |
| $DOI \cdot (DOI-\overline{DOI})^2$ | 0.02 | -0.06 | 0.51 | 1 | | | | | | | | | |
| RDI | 0.46 | 0.48 | -0.37 | -0.06 | 1 | | | | | | | | |
| MI | 0.12 | 0.10 | 0.00 | 0.11 | 0.28 | 1 | | | | | | | |
| CI | 0.19 | 0.07 | -0.01 | 0.08 | 0.21 | 0.05 | 1 | | | | | | |
| SC | -0.44 | -0.02 | -0.14 | -0.08 | -0.18 | -0.05 | 0.06 | 1 | | | | | |
| $(SC-\overline{SC})^2$ | 0.13 | 0.12 | -0.20 | -0.02 | -0.01 | 0.00 | 0.04 | 0.31 | 1 | | | | |
| ER | -0.43 | -0.09 | 0.13 | 0.08 | -0.38 | -0.38 | -0.08 | 0.37 | 0.01 | 1 | | | |
| BL | -0.16 | -0.04 | -0.03 | -0.05 | -0.04 | -0.11 | -0.02 | 0.05 | 0.02 | 0.16 | 1 | | |
| BM | 0.16 | 0.15 | -0.12 | -0.05 | 0.20 | 0.10 | 0.05 | 0.00 | 0.00 | -0.02 | 0.03 | 1 | |
| AG | -0.36 | -0.15 | -0.01 | -0.17 | -0.23 | -0.28 | -0.22 | 0.25 | 0.03 | 0.34 | 0.11 | -0.35 | 1 |

BL (-0.16), AG (-0.36) showed a negative correlation with CFP, while DOI (0.10), cubic term of DOI (0.02), RDI (0.46), MI (0.12), CI (0.19), square term of SC (0.13) and BM (0.16) There is a positive correlation with CFP, which indicates the relationship between the explanatory variables and the explained variables, and the actual situation (conditional analysis) still needs to be analyzed through the model., it can be found from Table 4 that there is no high correlation between variables, so this study can reasonably establish an econometric model.

## 4.1 Empirical Analysis

Information has been identified in this study DATA for the PANEL in front of the DATA, but the PANEL whether the DATA is suitable for using a PANEL DATA MODEL analysis, still need to decide, this research use Pooled regression, Pooled regression is weighted statistic and unweighted statistic, DATA, this study of weighted statistics $R2$ (0.59) is larger than unweighted statistics (0.47), and weighted statistic SSE (584) is smaller than unweighted statistics (764), Panel Data Analysis is suitable for the data in this study.

The PANEL DATA MODEL has a fixed effect and random effect. In this study, the chi-square test mentioned by Hausman (1978) was used for testing [41]. The result of the chi-square test was 100, $p \leq 0.001$, indicating that the fixed effect of this study's MODEL was the most appropriate. The analysis results are as follows.

$$CFP =$$
$$14.35 - 0.45DOI - 0.20DOI \cdot (DOI-\overline{DOI}) + 0.05DOI \cdot (DOI-\overline{DOI})^2 + 25.63RDI - 1.25MI$$
$$(13.25) \quad (-8.26) \quad (-5.93) \qquad\qquad (3.09) \qquad\qquad\qquad (11.29) \quad (-2.39)$$
$$*** \quad *** \quad *** \qquad\qquad *** \qquad\qquad\qquad *** \qquad **$$
$$+0.38CI - 0.55SC + 0.21(SC-\overline{SC})^2 - 0.17ER - 0.68PL + 0.35BM - 0.19AG$$
$$(2.80) \quad (-11.60) \quad (8.88) \qquad\quad (-3.08) \quad (-4.07) \quad (3.20) \quad (-4.20)$$
$$*** \quad *** \quad *** \qquad\quad *** \quad *** \quad *** \quad ***$$

Here, $R2 = 0.64$, indicating that there is no highly correlated problem between explanatory variables, where F-statistic = 42.06, and P-value<0.001. It also indicates that this model has sufficient explanatory power. Therefore, this study used this model for interpretation.

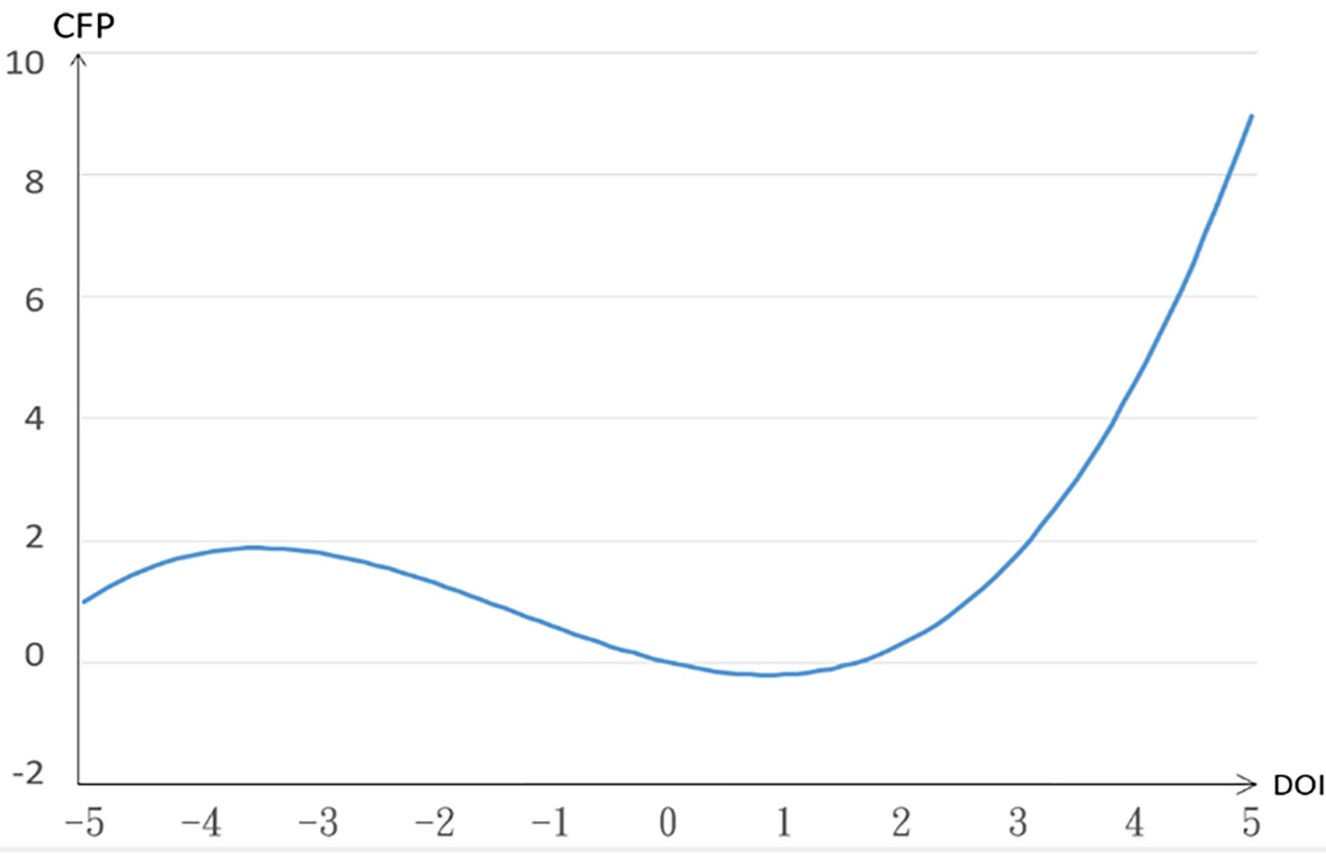

**Fig 1. The relationship between internationalization and corporate performance.**

From the model, it can be found that the degree of internationalization of the new retail industry presents a significant S-shape relationship (to the third power) with corporate performance. The relationship between the degree of internationalization and corporate performance is plotted through the numerical value, as shown in Fig 1.

From Fig 1, we find that the influence of internationalization on corporate performance has an S-shape. Early may accidentally order, don't need much cost, wait for the notice to the international market can be developed, a large number of input costs, lead to make ends meet, the adverse effects on the corporate performance, at the end of the market stable, overhead cost reduction, the increase of the accelerating type formed on corporate performance, but the pattern shows about the DOI = 3.3 (that is, about 4% of the export for sale in the domestic market), costs rise, let the degree of internationalization is affecting decreases until the DOI = 0.1 ~ 15 (1.1 ~ 4.48 times of export for sale in the domestic market). The degree of internationalization has a negative impact on corporate performance.

It is found that R&D intensity (25.63) and capital intensity (0.38) have a positive and significant impact on corporate performance, while marketing intensity (-1.45) has a negative and significant impact on corporate performance. This seems to suggest that the new retail industry should focus on researching how to reduce the benefits of logistics and advertising costs that far outweigh the benefits of direct input.

As for company size, this study also found that there is a threshold effect—that is, a positive U-shape. The relationship between company size and company performance is shown in Fig 2. To present economies of scale, this study finds that only 31.5 can have a positive impact on corporate performance.

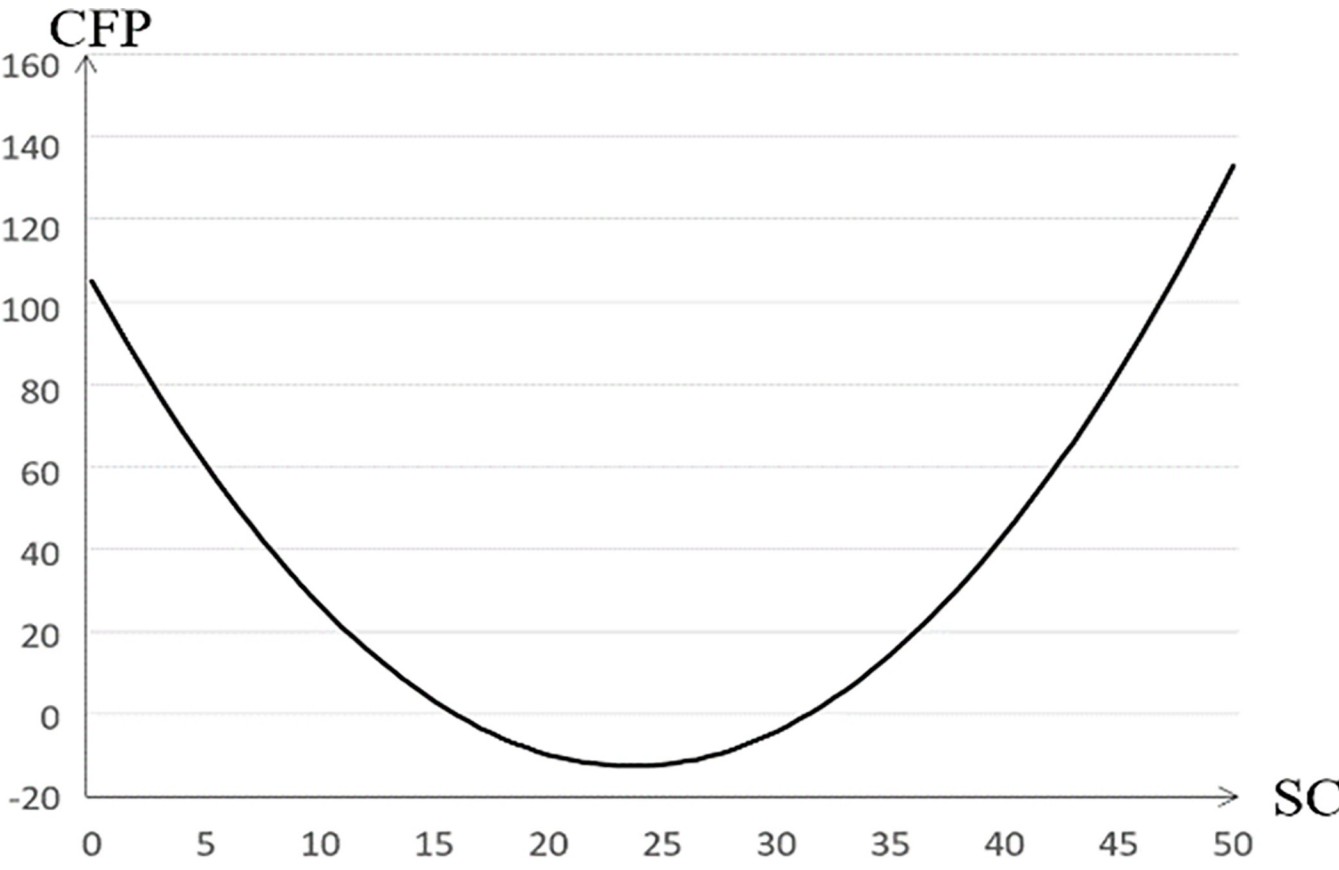

**Fig 2. The relationship between company size and company performance.**

The chairman and general manager (0.38) have a positive and significant impact on company performance, which also illustrates the importance of the integration of decision-making and implementation in the new retail industry. A longer listing time (-0.19) has a negative and significant influence on company performance, which also indicates that the longer the listing time is, the less beneficial popularity is to the new retail industry, and consumers attach more importance to the means of operators. The proportion of property right (-0.17) and the proportion of stock pledges of major shareholders (-0.68) have a negative and significant influence, as previously studied by scholars.

## 5. Discussion

For the new retail industry in China, we prove the following. The specificity of assets to a company on new retail performance is significant. In the whole process of our observation, a company's operations are more important, including providing a good service to consumers, or how to increase the value of its brand. All of these for companies are a plus. Through a series of the internationalization variable, we test the new retail companies to see if they invest a lot of cost in the process. Some things cannot illustrate the performance of a company. Thus, we observed some listed companies should have a land policy, in which senior directors can quickly see the relevant results of overall operations. The internationalization of the new retail industry is a big strategic direction for any company, and long-term planning is needed to have a positive overall development.

## 5.1 Theoretical implications

With the development of this novel research model, the theoretical contributions of this study to the literature are described as follows. We focus on the concept of the impact of new retail management on company performance management by conducting a literature review, analysis, and possible integration within this theoretical framework. Specifically, although the degree of internationalization and proprietary assets are central in the literature, notable gaps remain in understanding their effect on corporate performance and their impact on decision-making by managers and executives.

We have further developed the social capital theory, decision theory, expectations theory, and self-expectations theory, and the contribution of this paper to them is as follows. Our research provides a clearer understanding of the impact of internationalization of the new retail industry on corporate performance. The social capital theory mainly talks about the whole process of operating online and offline transactions. There is no real intangible asset value except for tangible real value. In terms of marketing intensity, such a measure is often overlooked financially and economically, resulting in a lot of intangibles in marketing intensity that can help create a brand.

We have broken the traditional enterprise take only the amount of real deal to represent the company's performance, some of the intangible value ignored on corporate performance, especially for many of China's retail industry e-commerce platform, although have the actual amount into the account, often ignore to the internationalization of professional knowledge and ability of the market, led to the company performance often neglected to intangible assets.

## 5.2 Managerial and practical implications

This research mainly neglects to look at the corporate performance from the perspective of the degree of internationalization in terms of improving the financial economy. The degree of internationalization is not easy to calculate and is applicable in terms of the overall financial situation. There is are interesting studies on the degree of internationalization within the One Belt and One Road policy of China. Many new retail industry companies have deployed assets overseas, with the ultimate purpose of influencing the degree of internationalization of the new retail industry on corporate performance. In this process, many enterprises ignore the management process. Many enterprises only see actual monetary figures in management, while ignoring intangible assets. It is suggested that future financial and accounting personnel should include and reevaluate this part of their assets.

## 6. Conclusions and future research

This study empirically proves the impact of the internationalization of the new retail industry on corporate performance and finds that R&D intensity and capital intensity of proprietary assets have a positive impact on corporate performance. Marketing intensity conversely has a negative impact. The main reason may be that marketing investment does not have a positive impact on the performance of a company immediately, because it takes a long time to accumulate and invest. Investment in intangible assets has become a new retail branding trend, but few people have paid attention to it and applied it. The new business model of the new retail industry can be promoted to the whole world through internationalization. A good business model is a very important indicator for China's new retail industry, and good products and business models need to be promoted to other parts of the world.

This research suggests that future studies can analyze different countries and markets. For example, big data analysis can be done on some countries in Southeast Asia or Europe on online and offline models. Taking Alibaba for example, it has begun to expand its business to

other countries in Southeast Asia. Another Chinese QQ also started to promote its business in Europe and the Netherlands through WeChat. Such data can be used to compare the impact of the internationalization of new retail industries in different countries on corporate performance. The degree of internationalization will lead to the enhancement of a company's proprietary assets and increase its corporate performance.

## Supporting information

**S1 Data.**
(XLSX)

## Author Contributions

**Conceptualization:** Li-Wei Lin.

**Formal analysis:** Shih-Yung Wei.

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
