## [Decision Letter · Decision Letter 0]

4 Mar 2022

PONE-D-21-28687The impact of internationalization of China's new retail industry on corporate performance - a moderating effect based on proprietary assetsPLOS ONE

Dear Dr. Lin,

Thank you for submitting your manuscript to PLOS ONE. After careful consideration, we feel that it has merit but does not fully meet PLOS ONE’s publication criteria as it currently stands. Therefore, we invite you to submit a revised version of the manuscript that addresses the points raised during the review process.

We look forward to receiving your revised manuscript.

Kind regards,

László Vasa, PhD

Academic Editor

PLOS ONE

Journal Requirements:

2. During our internal checks, the in-house editorial staff noted that you conducted  the phase 1 of your research in another country. Please check the relevant national regulations and laws applying to foreign researchers and state whether you obtained the required permits and approvals. Please address this in your ethics statement in both the manuscript and submission information. In addition, please ensure that you have suitably acknowledged the contributions of any local collaborators involved in this work in your authorship list and/or Acknowledgements. Authorship criteria is based on the International Committee of Medical Journal Editors (ICMJE) Uniform Requirements for Manuscripts Submitted to Biomedical Journals - for further information please see here: https://journals.plos.org/plosone/s/authorship

Reviewers' comments:

Reviewer's Responses to Questions

**Comments to the Author**

1. Is the manuscript technically sound, and do the data support the conclusions?

Reviewer #1: Yes

Reviewer #2: Partly

2. Has the statistical analysis been performed appropriately and rigorously? 

Reviewer #1: Yes

Reviewer #2: Yes

3. Have the authors made all data underlying the findings in their manuscript fully available?

Reviewer #1: Yes

Reviewer #2: No

4. Is the manuscript presented in an intelligible fashion and written in standard English?

Reviewer #1: Yes

Reviewer #2: No

5. Review Comments to the Author

Reviewer #1: I have enjoyed reading this paper. Introduction and literature review is well written. Methodology section is well designed. The authors had explained all the results in a critical fashion. This paper is suitable for publication.

Reviewer #2: The topic of the paper would be really interesting, however, there are some weaknesses.

Th abstract should be more clear and compact.

Introduction isn't complete enough,research questions and he context should be highlighted better.

Literature review is too short and only very few sources were processed in the paper.

The datasets are not provided, so I can not evaluate the validity of the results.

6. PLOS authors have the option to publish the peer review history of their article (what does this mean?). If published, this will include your full peer review and any attached files.

Reviewer #1: No

Reviewer #2: No

---

## [Author Response · Author response to Decision Letter 0]

4 Apr 2022

Thanks for your suggestion.

1. We have strengthened the literature sources for 2016-2021 in the literature.

2. Our research design is rigorous, including strict design in content, structure, and method, and has certain quality standards in publishing SSCI.

---

## [Editor Report · Decision Letter 1]

18 Apr 2022

The impact of the internationalization of China's new retail industry on corporate performance - A moderating effect based on proprietary assets

PONE-D-21-28687R1

Dear Dr. Lin,

We’re pleased to inform you that your manuscript has been judged scientifically suitable for publication and will be formally accepted for publication once it meets all outstanding technical requirements.

Kind regards,

László Vasa, PhD

Academic Editor

PLOS ONE
---

## [Editor Report · Acceptance letter]

19 May 2022

PONE-D-21-28687R1 

The impact of the internationalization of China's new retail industry on corporate performance - A moderating effect based on proprietary assets 

Dear Dr. Lin:

I'm pleased to inform you that your manuscript has been deemed suitable for publication in PLOS ONE. Congratulations! Your manuscript is now with our production department. 

Kind regards, 

on behalf of

Prof. Dr. László Vasa 

Academic Editor

PLOS ONE